



# Transport-driven aerosol differences above and below the canopy of a mixed deciduous forest

Alexander A.T. Bui[1], Henry W. Wallace[1,a], Sarah Kavassalis[2], Hariprasad D. Alwe[3], James H. Flynn[4], Matt H. Erickson[4,b], Sergio Alvarez[4], Dylan B. Millet[3], Allison L. Steiner[5], Robert J. Griffin[1,6]

[1] Department of Civil and Environmental Engineering, Rice University, Houston, TX, 77005, USA
[2] Department of Chemistry, University of Toronto, Toronto, ON, M5S 3H6, Canada
[3] Department of Soil, Water, and Climate, University of Minnesota, St. Paul, MN 55108, USA
[4] Department of Earth and Atmospheric Sciences, University of Houston, Houston, TX, 77204, USA
[5] Department of Climate and Space Sciences and Engineering, University of Michigan, Ann Arbor, MI, 48109, USA
[6] Department of Chemical and Biomolecular Engineering, Rice University, Houston, TX, 77005, USA
[a] now at: Washington State Department of Ecology, Lacey, WA, 98503, USA
[b] now at: TerraGraphics Environmental Engineering, Pasco, WA, 99301, USA

*Correspondence to*: Robert J. Griffin (rob.griffin@rice.edu)

**Abstract.** Exchanges of energy and mass between the surrounding air and plant surfaces occur below, within, and above a forest's vegetative canopy. The canopy also can lead to vertical gradients in light, trace gases, oxidant availability, turbulent mixing, and properties and concentrations of organic aerosols (OA). In this study, a high-resolution time-of-flight aerosol mass spectrometer is used to measure non-refractory submicron aerosol composition and concentration above (30m) and below (6m) a forest canopy in a mixed deciduous forest at the Program for Research on Oxidants: Photochemistry, Emissions, and Transport tower in northern Michigan during the summer of 2016. Three OA factors are resolved using positive matrix factorization: more-oxidized oxygenated organic aerosol (MO-OOA), isoprene-epoxydiol-derived organic aerosol (IEPOX-OA), and 91Fac (a factor characterized with a distinct fragment ion at $m/z$ 91) from both the above- and below-canopy inlets. MO-OOA was most strongly associated with long-range transport from more polluted regions to the south, while IEPOX-OA and 91Fac were associated with shorter-range transport and local oxidation chemistry. Overall vertical similarity in aerosol composition, degrees of oxidation, and diurnal profiles between the two inlets was observed throughout the campaign, which implies that rapid in-canopy transport of aerosols is efficient enough to cause relatively consistent vertical distributions of aerosols at this scale. However, four distinct vertical gradient episodes are identified for OA, with vertical concentration differences (above-canopy minus below-canopy concentrations) in total OA of up to 0.8 μg/m³. The magnitude of these differences correlated with concurrent vertical differences in either sulfate aerosol or ozone. These differences are likely driven by a combination of long-range transport mechanisms, canopy-scale mixing and local chemistry. These results emphasize the importance of including vertical and horizontal transport mechanisms when interpreting trace gas and aerosol data in forested environments.



## 1 Introduction

Aerosols play a key role in the energy balance of the Earth's climate system by scattering and absorbing incoming solar radiation and by impacting cloud lifetime and reflectivity (IPCC, 2007). These climatic effects depend strongly on the chemical speciation of the aerosol particles. Approximately 20-90% of submicron aerosol mass worldwide on average has been predicted to be organic material (Kanakidou et al., 2005), and this is supported by field studies in a variety of urban, urban downwind, and rural locations across the Northern Hemisphere (Jimenez et al., 2009; Zhang et al., 2007). The bulk of this organic material is thought to be secondary organic aerosol (SOA), which is formed in the atmosphere by the reaction of volatile organic compounds (VOCs) with oxidants such as hydroxyl radical (OH), ozone ($O_3$), nitrate radical, and chlorine atom, with the resulting products then partitioning to the particle phase.

Precursor VOCs that contribute to SOA formation are emitted from both anthropogenic and biogenic sources. Biogenic VOCs (BVOCs) primarily are emitted into the atmosphere from terrestrial vegetation, and on a global scale, emissions of BVOCs exceed those of anthropogenic VOCs (Fehsenfeld et al., 1992; Guenther et al., 2000, 1995). Major SOA precursor BVOCs include isoprene ($C_5H_8$) and terpenes. To date, more than 5000 terpene compounds have been identified, including monoterpenes ($C_{10}$), sesquiterpenes ($C_{15}$), and diterpenes ($C_{20}$) (Geron et al., 2000). However, factors such as the addition of functional groups (aldehydes, alcohols, carboxylic acids, alkyl nitrate, etc.) and the wide variety of possible reaction pathways rapidly increase the number of relevant atmospheric VOCs beyond what is initially emitted (Goldstein and Galbally, 2007).

If other loss processes, including deposition, were not considered, the final atmospheric fate of the carbon associated with BVOCs would be oxidation to carbon dioxide. However, partitioning of oxidation products to the aerosol phase as SOA interrupts this oxidation sequence. Partitioning of VOC oxidation products between the gas and aerosol phases depends on multiple factors such as the phase and concentration of pre-existing primary OA (POA) or SOA, particulate-phase SOA reactions, and the presence of aerosol liquid water (ALW) (Seinfeld and Pandis, 2006). As a result of the large number of precursor VOCs, the highly non-linear oxidation chemistry, and the presence of multiple aerosol phases in the atmosphere, SOA formation is complex and relatively poorly understood (Goldstein and Galbally, 2007).

The physical environment strongly impacts these chemical SOA formation processes. For example, in forested areas in which BVOC emissions are prevalent, the exchanges of energy and mass between the forest and the atmosphere are influenced by the forest's vegetation canopy. Absorption of light by a canopy can diminish the amount of radiation that is received below the canopy, influencing photolysis rates of photolabile species (Baldocchi et al., 1995; Brown et al., 2005; Fuentes et al., 2007; Makar et al., 2017; Schulze et al., 2017) and oxidant availability (Fuentes et al., 2007). Loss of BVOC oxidation products to deposition within the canopy also has been found to be an important factor in determining the oxidative capacity of a forested environment (Pugh et al., 2009).

Vertical transport, also influenced by the canopy, likewise impacts the concentrations of BVOC and SOA in forested environments. Roughness elements created by the leaves, branches, and stems in a dense vegetative canopy combined with above-canopy wind shear leads to coherent structures (Finnigan, 2000). These turbulent flow structures contribute to the fluxes of heat, energy, and matter in forest canopies (Thomas and Foken, 2007). The physical motion of coherent structures occurs via two main mechanisms: upward "bursts," in which air is ejected upward from the canopy into the atmosphere, and downward "sweeps," in which air is directed downward from the atmosphere into the canopy. Vertically resolved sonic anemometer measurements, which provide temperature and three-dimensional wind velocity components at each vertical measurement location, can be used to derive in-canopy mixing metrics.

The relative magnitudes of timescales for turbulent transport and chemical processing govern how trace compounds are distributed within the canopy (Baldocchi et al., 1995; Fuentes et al., 2007; Steiner et al., 2011). A modeling study by Gao et al. (1993) found that in-canopy chemical processing of isoprene occurs on a much longer timescale than turbulent transport, making in-canopy reactions less important compared to the turbulent transport and emission of isoprene in determining its



in-canopy concentrations. On the other hand, for compounds with an estimated chemical loss timescale that is roughly
equivalent to the timescales of turbulent transport (e.g., $O_3$-initiated oxidation of the sesquiterpene β-caryophyllene), rapid
in-canopy chemical loss could dominate (Stroud et al., 2005). Furthermore, partitioning to the aerosol-phase was inferred as
a potential reason for observations of decreased mixing ratios of oxidation products of very reactive BVOCs in a ponderosa
pine forest canopy (Holzinger et al., 2005).
Several studies have attempted to model vertical profiles of trace gases and aerosols in a forest canopy considering both
chemistry and turbulent transport. Bryan et al. (2012) found that forest canopy-atmosphere interactions were highly
sensitive to turbulent mixing parameterizations during a field campaign in northern Michigan. Differences in highly reactive
BVOCs and BVOC oxidation products have been estimated above and below the canopy in modeling and measurement
efforts (Alwe et al., 2019; Ashworth et al., 2015; Holzinger et al., 2005; Schulze et al., 2017; Stroud et al., 2005; Wolfe and
Thornton, 2011). Schulze et al. (2017) found that rapid through-canopy transport (minimum in-canopy residence time of 10
min) leads to relatively consistent simulated above- and below-canopy SOA composition and concentration.
Because in-canopy mixing plays a role in the vertical distribution of trace gases and aerosols in a forest canopy, vertical
differences in OA components and other inorganic aerosol species such as sulfate ($SO_4$) could be caused by the degree of
mixing between the above- and below-canopy environments. During the PROPHET-CABINEX 2009 campaign, the degree
of atmosphere-canopy coupling between the above-canopy atmosphere and the forest was analyzed by Steiner et al. (2011).
In this study, the degree of coupling was calculated using the ratio of the kinematic heat flux above the canopy to the
kinematic heat flux in the upper canopy. Opposing kinematic heat flux directions (negative ratios) imply that the below-
canopy environment is uncoupled from the atmosphere. In 2009, coupling conditions ranged between strong coupling, weak
coupling, and uncoupled. Uncoupled conditions occurred most commonly in the early morning hours between 4AM and
8AM local time. This set of hours represents approximately 30% of every day over the whole 2009 study period. This
suggests that early morning hours may contribute to more instances of uncoupled canopy-atmosphere conditions but that
coupling between the forest canopy and the atmosphere occurs a majority of the time (Steiner et al., 2011).
Past field studies, such as those above a tropical forest in Brazil and a temperate forest in California, found that in-
canopy SOA formation, deposition, and thermal gradient effects on gas-particle partitioning all influence net OA fluxes
(Farmer et al., 2013). In the same study, the authors found that oxygenated OA tended to deposit in the canopy, whereas the
forest canopy released less oxygenated OA. The source of OA fluxes between forests and the atmosphere has been
associated with vertical turbulent transport between the forest atmosphere and the surface layer directly above the canopy;
however, there have been few studies that have provided high temporal resolution measurements of both OA composition
throughout a forest canopy and canopy mixing strength. In a mixed forest in Ontario, Canada, Gordon et al. (2011) found
that the frequent occurrence of net upward aerosol flux was associated with decoupled canopy conditions where entrainment
of particle-free air from above the canopy created a positive flux above the forest. On the other hand, Whitehead et al.
(2010) found that the particle number concentration and submicron particle composition in the trunk space (i.e., below-
canopy) and above-canopy environments showed minimal variation with height during daytime due to stronger turbulence
and mixing conditions. In light of these limited studies, there is a need for additional measurements and data to inform the
exchange of aerosols between the forest and the atmosphere.
Recent work from the Program for Research on Oxidants and PHotochemistry, Emissions and Transport (PROPHET)
during the Atmospheric Measurements of Oxidants in Summer (AMOS) campaign in 2016 has indicated that flux of
isoprene and monoterpenes at the canopy-atmosphere boundary represents over half of the net carbon flux, while oxygenated
VOCs (OVOCs) constitute a majority of the species with detectable VOC fluxes (192 of the 236 species with identified
molecular formulas). The authors report that the observed and modelled net carbon flux during the campaign was upward
(canopy emission) during the campaign in 2016 (Millet et al., 2018). Vertical gradients of BVOCs in this mixed forest
environment vary greatly depending on canopy vegetation height, primary emission versus secondary production, and
diurnal variability (Alwe et al., 2019). Correlation analysis of VOC vertical gradients suggests that formic acid (HCOOH)
originates from a secondary photochemical production. During the same campaign, distinct sample-to-sample variability in





the molecular-level aerosol composition (73 +/- 8%) was observed despite less variability in elemental composition of the
bulk OA, indicating the chemical complexity of functionalized OA at the site (Ditto et al., 2018).
Here, measurements using an Aerodyne high-resolution time-of-flight aerosol mass spectrometer (HR-ToF-AMS) are
used to characterize ambient aerosol above and below the canopy in a mixed, deciduous forest during the PROPHET-AMOS
campaign in 2016. These measurements provide data for validation of forest canopy-atmosphere exchange models and
allow an assessment of turbulent transport in the forest canopy and the resulting impacts on above- and below-canopy SOA
composition and concentration. These data also could assist in the determination of whether or not SOA is forming within
the forest canopy.

## 129   2 Methods

### 130   2.1 Site description

Measurements were made during the PROPHET-AMOS 2016 campaign from 1 July – 31 July 2016. The PROPHET
site (45.55° N, 84.78° W) is situated in a temperate, mixed deciduous forest in the northern portion of Michigan's Lower
Peninsula at the University of Michigan Biological Station (UMBS). The surrounding forest consists of aspens (60.9%),
northern hardwoods (16.6%; maple, beech, birch, ash, and hemlocks), upland conifers (13.3%; white and red pines), non-
forest cover types (7.6%; bracken ferns/grass/developed/road), and northern red oak (1.6%) (Cooper et al., 2001; Bergen and
Dronova, 2007). The forests in this region are currently undergoing succession, where the dominant aspens have matured
and are now being replaced by northern hardwoods and pines (Bergen and Dronova, 2007). It is expected that this new
forest composition will shift BVOC emissions from an isoprene-dominated environment to one that is more influenced by
monoterpenes (Toma and Bertman, 2012). The height of the forest canopy varies, but the mean canopy height is
approximately 22.5 meters (VanReken et al., 2015). The site's physical layout and site meteorology have been described
elsewhere (Carroll et al., 2001; Cooper et al., 2001). The PROPHET site features a 31-m scaffolding tower, allowing
measurements to be made at variable heights within and above the vegetation canopy.
Due to the sparse surrounding population, the PROPHET site at UMBS has minimal local anthropogenic influences.
The closest major urban centers include Detroit, Michigan (350 km to the southeast, population: 672,795), Milwaukee,
Wisconsin (350 km to the southwest, population: 595,047), and Chicago, Illinois (450 km to the southwest, population:
2,704,958) (Carroll et al., 2001). The closest nearby towns include Pellston, Michigan (5 km to the west, population: 828),
Petoskey, Michigan (30 km to the southwest, population: 5,749), and Cheboygan, Michigan (30 km to the northeast,
population: 4,726). Population totals are based on 2016 population estimates for cities and towns in the United States (US)
from the US Census Bureau (US Census Bureau, 2018).
The mean temperature during the campaign was 20.6 +/- 4.6 °C (mean +/- one standard deviation), which is 3-4°C
warmer than the mean temperature during the 2009 PROPHET-CABINEX campaign (16.9 °C) (VanReken et al., 2015).
Temperature conditions during this study are consistent with mean summertime temperatures from studies in 2008 and 2010
(Toma and Bertman, 2012) and with historical mean temperature data for the month of July from the Pellston Regional
Airport (Figure S1 in the Supplemental Information (SI)). The mean relative humidity (RH) during the campaign was 73.8
+/- 17.5% , which is similar to conditions during the CABINEX campaign (74.5 +/- 17.5%) (VanReken et al., 2015). Winds
recorded at the top of the PROPHET tower originated mostly from the west, southwest, and northwest (as shown in Figure
S2 in the SI). The historical average precipitation according to the National Atmospheric Deposition Program (NADP)
National Trends Network for the month of July from 1979-2015 for UMBS is 69.0 +/- 37.8 mm. During the PROPHET-
AMOS 2016 campaign, the total accumulated precipitation was 79.0 mm, indicating that precipitation at the site during the
campaign was within one standard deviation of the historical July average (NADP, 2016).



## 2.2 Instrumentation and sampling

A summary of the trace-gas measurements, instrumentation, and meteorological parameters on board the University of Houston/Rice University Mobile Air Quality Laboratory (MAQL) are summarized in Table S1 of the SI. Details and operation of the MAQL have been described previously in the literature (Leong et al., 2017; Wallace et al., 2018). The MAQL was situated approximately 10 meters to the east of the PROPHET tower. A photograph of the MAQL in stationary sampling mode is shown in Figure S3 in the SI. Table S2 in the SI lists the measurements, measurement methods, and sampling heights of other participating institutions at PROPHET-AMOS 2016 during the campaign. Measurements are reported in local time (Eastern Daylight Time (EDT)) during the campaign.

### 2.2.1 Trace gases

Below-canopy trace gases including nitric oxide (NO), nitrogen dioxide ($NO_2$), total reactive nitrogen ($NO_y$), $O_3$, carbon monoxide (CO), and sulfur dioxide ($SO_2$) were measured from a common inlet on the sampling arm of the MAQL. Reported values of nitrogen oxides ($NO_x$) represent the sum of NO and $NO_2$, while reported values of $NO_y$ include $NO_x$ and its reservoir species. These measurements were taken at a height of 6 m above ground level.

### 2.2.2 HR-ToF-AMS

The HR-ToF-AMS (Aerodyne Research Inc., USA) was used to measure non-refractory submicron particulate matter ($NR-PM_1$) from the MAQL; measured composition includes OA, $SO_4$, nitrate ($NO_3$), ammonium ($NH_4$), and chloride (Chl). Detailed descriptions of the operation and principles of the HR-ToF-AMS have been provided elsewhere (DeCarlo et al., 2007). In brief, particles are sampled through a 100-micron critical orifice and are focused into a particle beam using an aerodynamic lens. After traversing a vacuum chamber, the particles in the beam impact onto a tungsten vaporizer heated to 600°C. The vapors formed are ionized by electron impact ionization at 70 eV. The resulting ions are detected using a high-resolution time-of-flight mass spectrometer. The HR-ToF-AMS was operated in a high mass sensitivity mode, referred to as V-mode. Ionization efficiency (IE) calibrations were performed at the beginning and end of the campaign using monodisperse 300 nanometer $NH_4NO_3$ particles. Gas-phase interferences were subtracted from the data based on the observed signal when ambient air was sampled through a filter. Filter zeros were run each day at varying times of day.

Sampling was performed from the raised 6-m inlet on the common sampling arm of the MAQL (below canopy) and from a 30-m inlet on the PROPHET tower (above canopy). Copper tubing was used for the sampling inlets, and each inlet was fitted with cyclones to remove particles larger than 2.5 microns in diameter. Prior to the HR-ToF-AMS critical orifice, air was sampled through a nafion dryer to dry the sample flow. A three-way valve was used to alternate HR-ToF-AMS sampling between the above- and below-canopy inlets at 10-minute intervals.

### 2.2.3 PTR-QiToF

Above-canopy mixing ratios and fluxes, along with in-canopy vertical gradients, were measured for a wide array of VOCs during PROPHET-AMOS 2016 by Proton Transfer Reaction-Quadrupole interface Time-Of-Flight mass spectrometry (PTR-QiToF). A detailed description of the sampling configuration, calibration and zeroing procedures, humidity corrections, and instrumental performance during the campaign is provided by Alwe et al. (2019) and Millet et al. (2018). Briefly, six identical 45-m inlet lines (0.5" OD/0.375" ID PFA, each heated to 50°C) were installed on the PROPHET tower to sample from 34 m, 21 m, 17 m, 13 m, 9 m, and 5 m above ground level. Sample flow was maintained at ~40 standard L/min (SLM) for the 34 m inlet line and at >5-10 SLM for the others. Each hour, 30 min was spent sampling from the 34 m inlet to quantify above-canopy VOC mixing ratios and fluxes. The remainder of each hour was spent characterizing in-canopy vertical gradients by sequentially sampling from the other inlets (for 5 min apiece) followed by a 5-min instrumental blank.



### 2.3 HR-ToF-AMS data and positive matrix factorization analysis

Data analysis for HR-ToF-AMS data was performed in Igor Pro v6.37 using the SQUIRREL v1.57 (SeQUential Igor data RetRiEvaL) and PIKA v1.16 (Peak Integration by Key Analysis) analysis toolkits. High-resolution mass spectral fitting was performed on HR-ToF-AMS V-mode data. Ratios such as the oxygen-to-carbon elemental ratio (O:C) and hydrogen-to-carbon elemental ratio (H:C) were determined according to the "improved-ambient" (IA) method (Canagaratna et al., 2015). The IA method uses specific ion fragments to correct for compositional biases, and this method has been shown to calculate accurately the elemental ratios of organic laboratory standards that are more representative of oxidized, ambient OA species (Canagaratna et al., 2015). The default values for relative IE were used for each of the following species: OA (1.4), $SO_4$ (1.2), $NH_4$ (4), $NO_3$ (1.1), and Chl (1.3), where values in parentheses refer to the ratio of the IE of the given species with respect to the value of IE of $NO_3$ obtained during routine IE calibrations using $NH_4NO_3$.

To account for the effects of aerosol composition on the transmission efficiency of aerosols to the detection region of the HR-ToF-AMS, a chemical composition-dependent (and therefore time-dependent) collection efficiency (CDCE) was applied to the HR-ToF-AMS data (Middlebrook et al., 2012), which led to an campaign average CDCE of 0.77 +/- 0.18. During the campaign, the HR-ToF-AMS time resolution was 40 seconds from 2 July 02:00 to 19 July 10:00 EDT, after which the HR-ToF-AMS time resolution was changed to 30 seconds between 22 July 11:00 and 31 July 15:00 EDT (due to instrumental changes after an HR-ToF-AMS power supply failure). Calculated detection limits of NR-PM$_1$ species are included in Table S3.

Positive matrix factorization (PMF) is a mathematical model in which measured data are decomposed into a combination of factors that have varying contributions throughout a time series. Here, the PMF model has been applied to HR-ToF-AMS data to retrieve OA factors that contain information regarding OA sources, chemical properties, and/or atmospheric processing (Jimenez et al., 2009; Ulbrich et al., 2009; Zhang et al., 2011). Subtypes of OA extracted from OA mass spectra using PMF often include (but are not limited to): more-oxidized oxygenated OA (MO-OOA), less-oxidized oxygenated OA (LO-OOA), hydrocarbon-like OA (HOA), biomass burning organic aerosol (BBOA), and cooking organic aerosol (COA) (Cubison et al., 2011; Jimenez et al., 2009; Mohr et al., 2012; Zhang et al., 2007, 2005). The OOA-related factors are generally considered to be associated with SOA, while subtypes such as HOA, BBOA, and COA are presumed to correspond to POA. Factors associated with isoprene-derived epoxydiol OA (IEPOX-OA) have also been identified using PMF using enhanced signals at m/z 82 in their OA mass spectra (Hu et al., 2015).

Factorization of HR-ToF-AMS data follows the matrix form:

$$X = GF + E \tag{1}$$

where $X$ is a $m \times n$ matrix in which rows are the measured mass spectra at each time interval and columns are the time-varying signals of each sampled mass-to-charge ratio (m/z). $G$ is a $m \times p$ matrix in which rows are the concentration time series for a given factor, and the number of factors (columns) in the solution is represented by $p$. $F$ is a $p \times n$ matrix in which $p$ rows are the mass spectral profiles for a given factor. Finally, $E$ is a $m \times n$ matrix and contains the residuals not fit by the model at each time interval and sampled m/z. Using a least-squares algorithm, the values of $G$ and $F$ are determined by minimizing $E$ using a quality of fit parameter ($Q$) defined as the squared sum of the scaled residuals:

$$Q = \sum_{i=1}^{m} \sum_{j=1}^{n} \left( \frac{e_{ij}}{\sigma_{ij}} \right)^2 \tag{2}$$

where $\sigma_{ij}$ is the estimated standard deviation of the points in the data matrix, $X$, and $e_{ij}$ is an element in in the model residual matrix, $E$. Scaling in $Q$ is calculated using reduced weights towards outliers, thus allowing for weighting of the input data by their level of measurement certainty (standard deviations in the data matrix, $X$) (Paatero and Tapper, 1994; Ulbrich et al., 2009). The number of factors is generally unknown a priori and is determined based on the interpretation of PMF results. The selection of the number of factors can be aided using mathematical metrics, including $Q/Q_{exp}$, where $Q_{exp}$ represents the degrees of freedom in the dataset ($Q_{exp} = mn - p(m + n)$). As values of $Q/Q_{exp}$ approach 1, the appropriate number of modeled factors is determined.





In this study, PMF is applied separately to the above- and below- canopy HR-ToF-AMS OA high-resolution mass
spectra between m/z 12 and m/z 115 using the generic 64-bit PMF2 v4.2 algorithm running in robust mode with a model
error set to zero (Paatero, 1997; Paatero and Tapper, 1994; Ulbrich et al., 2009). The default PMF2 convergence criteria
were used. The PMF Evaluation Tool v2.08 (PET) in Igor Pro v6.37 was used to treat the OA mass spectra error matrix,
evaluate PMF output, and examine model statistics (Ulbrich et al., 2009). Organic isotopes were excluded from PMF
analysis because isotope signals are scaled from their parent ions rather than being measured directly. A minimum error
threshold was applied to the error matrix where any error values falling below this threshold are replaced. Variables (time
series of m/z values) with a signal-to-noise ratio (SNR) less than 0.2 were removed, and variables with a SNR less than 2
were downweighted by a factor of 2. The error values for fragments such as $O^+$, $HO^+$, $H_2O^+$, and $CO_2^+$ were also
downweighted to avoid providing additional weight to the strong signal attributed to m/z 44 in the default fragmentation
table. To decrease the weight of each of these fragments, the error values for each of these fragment ions are all multiplied
by a factor of 2 to appropriately downweight the m/z values related to m/z 44. Additionally, the error of the $CHO^+$ fragment
was downweighted by multiplying the error value for this fragment by a factor of 4. Downweighting of $CHO^+$ was
performed because m/z 29 is a combination of signals from the $CHO^+$ ion (m/z 29.0027) and a $N_2$ isotope ion (j15NN, m/z
29.0032). The close proximity of the $CHO^+$ and j15NN fragments (< 0.001) using PIKA v1.16 likely causes the $CHO^+$ error
to be underestimated (Xu et al., 2015).
The optimal PMF solution for the OA data was determined by examining the following: (1) time series and mass spectra
model residuals, (2) interpretability of factor diurnal variability, (3) correlations between factor time series and time series of
external data, individual values, or tracers, (4) factor mass spectral characteristics, and (5) reductions in $Q/Q_{exp}$. The
rotational ambiguity for each solution was explored by running PMF under a number of different FPEAK values, ranging
from -1.0 to 1.0 in increments of 0.2. A change to the FPEAK parameter explores the different linear transformations (also
referred to as "rotations") of a given solution that result in identical fits to the data (Ulbrich et al., 2009). The robustness of
each solution was evaluated by initializing the PMF model for a number of different starting points (or SEED values),
ranging from 0 to 50 in increments of 1. In addition to the FPEAK and SEED analysis performed on the optimal PMF
solution, the quantitative uncertainty of the solution is performed on the original m/z and time series data using
bootstrapping analysis, where 100 PMF model runs are executed with replacement of the mass spectra. Variations (1σ) of
the average factor mass spectra at each m/z and average factor time series are used to assess the robustness of the optimal
PMF solution (Ulbrich et al., 2009).
The results of the separate PMF analyses on above- and below-canopy OA are included in the SI. For the above-canopy
OA data, a summary of the PMF factor selection (Table S4), factor time series correlations with external data (Table S5),
factor mass spectra correlations with reference mass spectra (Table S6), time series of PMF model residuals (Figure S4), and
mass spectra and time series for possible two- to five-factor PMF solutions (Figure S5 - Figure S8) are shown in the SI.
VOCs measured above-canopy via PTR-QiToF at the top of the PROPHET tower are defined in Table S7, and factor time
series correlations with the time series of these VOCs are shown in Table S8. PMF diagnostics, such as the mass spectra and
time series correlation amongst factors (Figure S9), FPEAK and SEED diagnostic plots (Figure S10), results of the FPEAK
analysis (Table S9), model residual diagnostic plots (Figure S11), and results from bootstrapping analysis (Figure S12) are
shown in the SI. Finally, Figure S13 displays the time series and high-resolution mass spectra of the optimal solution for the
above-canopy OA dataset.
For the below-canopy OA data, a summary of the PMF factor selection (Table S10), factor time series correlations with
external data (Table S11), factor mass spectra correlations with reference mass spectra (Table S12), mass spectra and time
series of possible two- to five-factor PMF solutions (Figure S14 - Figure S17), factor time series correlations with VOCs
measured via PTR-QiToF at the 34-m inlet on the PROPHET tower (Table S13), mass spectra and time series correlations
amongst factors (Figure S18), FPEAK and SEED diagnostic plots (Figure S19), results from FPEAK analysis (Table S14
and Table S15), model residual diagnostic plots (Figure S20), and results from bootstrapping analysis (Figure S21) are
shown in the SI. Finally, the time series and high-resolution mass spectra of the optimal three-factor solution for below-
canopy OA are shown in Figure S22.



### 2.4 HYSPLIT backward trajectory analysis

#### 2.4.1 Trajectory cluster analysis

Backward-trajectories are used in this study to determine the origin of air masses arriving at the field site using the Hybrid Single-Particle Lagrangian Integrated Trajectory (HYSPLIT) model (Draxler and Hess, 1998; Stein et al., 2015). Meteorological data from the US Eta data assimilation system archive at 40-km spatial resolution (EDAS40) are used for HYSPLIT trajectory calculations. The EDAS40 data output is constructed using forecasted data from the Eta model, which utilizes observations from surface, aircraft, and satellite data to predict meteorological parameters such as pressure, wind speed, and wind direction (Cooper et al., 2001).

In order to assess the influence of air mass histories on aerosols at each site, a cluster analysis was performed on backward trajectories using MeteoInfo v1.4.9R2 and the TrajStat v1.4.4R8 package (Wang, 2014; Wang et al., 2009). The angle distance clustering type is used in this study and calculates the angular distance between two backward trajectories as seen from the site, using methods outlined in Sirois and Bottenheim (1995). The number of suitable clusters is chosen based on the slope of the percentage change in total spatial variation versus number of clusters and a visual inspection of the mean trajectories of the cluster numbers.

#### 2.4.2 Weighted potential source contribution function (WPSCF) analysis

In addition to a backward-trajectory cluster analysis performed for bulk aerosol properties and gas-phase species, two-day HYSPLIT backward-trajectories initiated from the PROPHET site at 500 m above ground level are used in a weighted potential source contribution function (WPSCF) analysis for OA factors. WPSCF analysis is performed in MeteoInfo v1.4.9R2 using the TrajStat v1.4.4R8 package and results are plotted using ESRI's ArcMap v10.1 (Wang, 2014; Wang et al., 2009). Similar PSCF analyses using backward-trajectories have been performed previously using aerosol properties (Bondy et al., 2017; Chang et al., 2017; Polissar, 1999; Schulze et al., 2018).

The number of backward-trajectory endpoints falling within a given grid cell with coordinates $(i, j)$ is defined as $n_{ij}$. The number of instances in which backward-trajectories ending at a given grid cell have a value (i.e., OA factor mass concentration) higher than an arbitrarily set criterion value is defined as $m_{ij}$. The PSCF value for a cell at location $(i, j)$ is then defined as follows:

$$PSCF_{ij} = \frac{m_{ij}}{n_{ij}} \tag{3}$$

When calculating values of PSCF, some grid cells will contain only a small number of backward-trajectory endpoints. In order to reduce the high uncertainties related to a limited number of endpoints falling within a grid cell in PSCF analysis, a weighting function is applied to the trajectory numbers following the methods of Polissar et al. (2001):

$$W_{ij} = \begin{cases} 1.00 & 80 < n_{ij} \\ 0.70 & 20 < n_{ij} \leq 80 \\ 0.42 & 10 < n_{ij} \leq 20 \\ 0.05 & n_{ij} \leq 10 \end{cases} \tag{4}$$

In this study, the domain of analysis is set to the geographical extents of the two-day HYSPLIT backward trajectories initiated from the PROPHET site. A grid cell size of 0.5° by 0.5° is used. Median values of the OA factors are used as the arbitrary criterion values. Overall, the WPSCF analysis allows for an identification of potential source areas, where higher WPSCF values within a region indicate a higher likelihood that this region results in observed values higher than the criterion values.



### 2.5 Sonic anemometer data-processing

Turbulence measurements during the campaign were obtained from five sonic anemometers installed on the PROPHET tower at the following measurement heights: 34m (CSAT 3B, Campbell Scientific Inc.), 29m (81000, RM Young), 21m (CSAT 3, Campbell Scientific Inc.), 13m (CSAT 3, Campbell Scientific Inc.), and 5m (CSAT 3, Campbell Scientific Inc.). The sonic anemometer at 34 m was operated continuously during the campaign while data are only available from the lower heights from July 9$^{th}$, 2016 – July 29$^{th}$, 2016. High-frequency data are de-spiked (data points outside of 3.5-standard deviations are removed) and then separated into 30-minute windows to apply a tilt correction such that the x-axis is rotated into the direction of the mean wind velocity (Foken, 2008). Reynolds decomposition is then applied to the three-dimensional wind components $(u, v, w)$, so each variable (e.g., $u$) is separated into its mean ($\bar{u}$) and fluctuating component ($u'$). The friction velocity, u*, is defined then as

$$u*^2 = -\overline{u'w'} \tag{5}$$

Any 30-minute periods that experienced rain (as measured by the rain-gauge at the UMBS AmeriFlux tower), weak winds (winds less than 0.5 m s$^{-1}$ at the top sonic anemometer), or wind directed through the tower were excluded due to potential interference.

## 3 Results and discussion

### 3.1 Backward-trajectory cluster analysis

Two-day backward trajectories are initiated from the PROPHET site and calculated at one-hour intervals from the beginning to the end of the campaign (1 July 00:00 – 31 July 21:00 EDT) at 500 meters above ground level, an elevation selected to be within the boundary layer during the day and to avoid trajectory interaction with the surface. A total of 742 two-day backward-trajectories are calculated over the course of the campaign. Cluster analysis resulted in three directional clusters: southerly (299 of 742), northeasterly (192 of 742), and northwesterly (251 of 742) (as shown in Figure S23 in the SI as Cluster 1, 2, and 3 respectively). Further description of cluster number selection is discussed in Figure S24 in the SI. Similar to the air mass history analyses at the PROPHET site in Cooper et al. (2001) and VanReken et al. (2015), eight-hour transitional periods between backward-trajectory classifications were removed from analysis because it is likely that the chemical species from these different air masses are mixed. In total, transitional periods composed 22% of the total number of backward-trajectories (165 of 742 total), and the remaining 577 are used for further analysis. Mean values of anthropogenically-influenced species such as SO$_4$ and benzene were not statistically significantly different ($p < 0.01$) between trajectories from northeasterly and northwesterly clusters, so trajectories from these two clusters are grouped to represent a "northerly" air mass type. During the PROPHET-AMOS study, northerly transport occurred during 60% of the study period (341 of 577), while southerly transport occurred 40% of the time (236 of 577). This type of air-mass classification is consistent with previous summertime studies at the PROPHET site where northerly transport occurred 44% (1998), 60% (2009), and 57% (2014) of the time and southerly transport occurred 24% (1998), 29% (2009), and 43% (2014) of the time (Cooper et al., 2001; Gunsch et al., 2017; VanReken et al., 2015).

### 3.2 Non-refractory submicron time series and bulk chemical composition

Figure 1 shows the time series of the mass concentrations of OA, SO$_4$, NH$_4$, NO$_3$, and Chl as measured by the HR-ToF-AMS. Mass concentrations plotted in Figure 1 represent time series data from both the above- and below-canopy sampling inlets. The average total NR-PM$_1$ (sum of mass concentrations of OA, SO$_4$, NH$_4$, NO$_3$, and Chl) is 2.3 +/- 1.5 µg m$^{-3}$. High NR-PM$_1$ concentration episodes (7/3-7/7 and 7/11-7/14) are strongly influenced by southerly air masses advecting

**Figure 1: Overview of time series of the following from top to bottom: temperature (T) and RH; pressure (Press.) and precipitation (Precip.); wind direction (WD) and wind speed (WS); CO and O₃; SO₂, NOₓ, and NOᵧ; isoprene (Isop.) and total monoterpenes (Monot.); particulate OA (green), NO₃ (blue), SO₄ (red), NH₄ (orange), and Chl (purple); fraction of species to total NR-PM₁, OA factors derived from PMF, and fractional contribution of OA factors to total OA. OA factors are colored as follows: MO-OOA (black), IEPOX-OA (red), and 91Fac (green). Precipitation data are provided from the NADP site in Cheboygan County, MI. Trace gas data (CO, O₃, SO₂, NOₓ, and NOᵧ) are measured from the 6-m inlet on the MAQL. Meteorological data (T, RH, Press., WD, and WS) and VOC data (Isop. and Monot.) are measured from the 34-m inlet on the PROPHET tower.**

to the site, as confirmed by the HYSPLIT backward-trajectory clusters shown in Figure S23 in the SI. Northerly backward-trajectories originated over clean, remote areas in Canada, while southerly backward-trajectories originated over more anthropogenically influenced areas. OA is the dominant NR-PM₁ component over the entire campaign, representing approximately 84.2% of the NR-PM₁ mass. SO₄ contributes the second highest average mass fraction to NR-PM₁ (10.7%)





followed by NH$_4$ (3.1%), NO$_3$ (1.6%), and Chl (0.4%). During periods of northerly flow, OA represents 89.5% of the
average NR-PM$_1$ mass, while SO$_4$, NH$_4$, NO$_3$ and Chl represent 6.9%, 1.9%, 1.3%, and 0.4%, respectively. Periods of
southerly flow decrease the relative contribution of OA to 75.5% and Chl to 0.3% while increasing the relative contribution
of SO$_4$ to 16.8%, NH$_4$ to 5.6%, and NO$_3$ to 1.3%. The increased fractional contribution of OA during periods of northerly-
originating air is consistent with results from VanReken et al. (2015), who found that water-soluble organics dominated
aerosol mass during periods of "clean" northerly flow at the PROPHET site. Sheesley et al. (2004) also found the aerosol
organic carbon composition at a remote site in the upper peninsula of Michigan (approximately 100 miles from the
PROPHET site) was greatly influenced by the source region of the air parcel. This study found that both stagnant and
northerly air parcels contained higher concentrations of pinonic acid and limited amounts of primary emission tracer
compounds, while anthropogenically influenced air parcels from the south and northwest contained higher concentrations of
aromatic and aliphatic dicarboxylic acids.
Diurnal plots for OA, SO$_4$, NH$_4$, NO$_3$, O:C, and H:C are shown in Figure S25 in the SI. The diurnal profiles of OA,
SO$_4$, and NH$_4$ are all relatively flat and do not have a clear diurnal trend. The lack of clear diurnal variations for SO$_4$ is
consistent with regional transport as the source of SO$_4$ during this campaign. The diurnal variations of NO$_3$ show increases
in the morning, with a maximum around approximately 10:00 local time, and lower concentrations in the afternoon.
Using the backward-trajectory clustering results described in Section 3.1, mean values of NR-PM$_1$, NR-PM$_1$ species, OA
elemental ratios, meteorological parameters, and trace gases for the entire campaign, northerly, and southerly backward-
trajectory clusters are shown in Table 1. Overall, relative to northerly air masses, southerly air masses were found to be
warmer (~2 °C), more humid (~5%), and associated with higher concentrations of NO$_x$, NO$_y$, O$_3$, and benzene. Furthermore,
on average, southerly air masses had higher concentrations of NR-PM$_1$, OA, SO$_4$, NH$_4$, and NO$_3$. Higher levels of OA
oxidation (8% difference) based on O:C are also observed during periods of southerly flow (O:C = 0.69 versus 0.75 for
northerly versus southerly flow, respectively). An additional metric of oxidation, the oxidation state of carbon (OSc),
indicates that the degree of oxidation is higher during periods of southerly flow (OSc = -0.12 versus 0.06 for northerly versus
southerly flow, respectively, where OSc = 2 * O:C – H:C) (Kroll et al., 2011). The factor of 5 differences between northerly
and southerly SO$_4$ is consistent with the increased influence of SO$_2$ point sources from electric-generating units south of the
site in Ohio, Indiana, and Illinois. Overall, these observations demonstrate that anthropogenic influence from southerly flow
directly affects NR-PM$_1$ mass concentrations, NR-PM$_1$ composition, the degree of OA oxidation, and trace gas mixing ratios
at this site. Results are in agreement with VanReken et al. (2015), where southerly air masses or those "anthropogenically
impacted" were found to contain higher aerosol loadings (in terms of aerosol volume, particle number, and median particle
diameter) and hygroscopicity, along with increased trace gases abundances, as compared to northerly air masses or "clean"
regimes.
Oxygen-containing ion families (C$_x$H$_y$O$_{z\geq1}^+$) represent over 50% of the campaign-averaged OA high-resolution mass
spectrum, and this high degree of oxygenation is reflected in an average O:C ratio of 0.71 +/- 0.08 and an average H:C ratio
of 1.49 +/- 0.06. A distinct peak at m/z 44 in the average mass spectrum accounts for 14.1% of the total OA. The peak at
m/z 44 is mainly composed of the CO$_2^+$ ion (96.0% of m/z 44). The ratio of m/z 44 to the total signal in the organic mass
spectrum (f$_{44}$), a surrogate for O:C and an indicator for photochemical aging, in this study is 0.14. Together with the average
O:C ratio (0.71) observed in this study, these values are consistent with OOA observed across AMS datasets (Jimenez et al.,
2009; Ng et al., 2010). Other prominent ions in the campaign-averaged high-resolution mass spectrum include m/z 55, 82,
and 91. Fragments at m/z 55 represent 2.4% of the total OA, and are representative of both oxygenated and hydrocarbon
fragments, such as C$_3$H$_3$O$^+$ (50.2% of m/z 55) and C$_4$H$_7^+$ (42.9% of m/z 55). The possible OA sources leading to increased
signal at m/z 82 (0.46% of total OA) and 91 (0.62% of total OA) will be discussed in the following section. Figure S25 in
the SI shows the diurnal variations of O:C and H:C, indicating relatively stable diurnal cycles. Increases in H:C are observed
starting at 9:00 before reaching a maximum at midday (13:00) followed by a slow decrease between the hours of 13:00 and
20:00, while an opposite pattern is observed for mean values of O:C.





**Table 1: Campaign-averaged values (+/- one standard deviation from the mean) of mass concentrations of NR-PM$_1$, VOC and**
**trace gas mixing ratios, and meteorological parameters, as well as hourly-averaged values associated with northerly and southerly**
**backward-trajectory clusters. Northerly and southerly air masses are defined in Section 3.1 using a cluster analysis of HYSPLIT**
**two-day backward trajectories.**

| Parameter | Campaign | Northerly | Southerly |
|---|---|---|---|
| NR-PM$_1$ ($\mu$g m$^{-3}$) | 2.3 ± 1.5 | 1.8 ± 0.7 | 3.6 ± 2.0 |
| OA ($\mu$g m$^{-3}$) | 1.9 ± 1.0 | 1.6 ± 0.6 | 2.6 ± 1.3 |
| SO$_4$ ($\mu$g m$^{-3}$) | 0.3 ± 0.4 | 0.1 ± 0.1 | 0.7 ± 0.6 |
| NH$_4$ ($\mu$g m$^{-3}$) | 0.1 ± 0.1 | 0.04 ± 0.04 | 0.2 ± 0.2 |
| NO$_3$ ($\mu$g m$^{-3}$) | 0.04 ± 0.03 | 0.02 ± 0.01 | 0.06 ± 0.05 |
| Chl ($\mu$g m$^{-3}$) | 0.01 ± 0.00 | 0.01 ± 0.00 | 0.01 ± 0.00 |
| O:C | 0.7 ± 0.1 | 0.7 ± 0.1 | 0.8 ± 0.1 |
| H:C | 1.5 ± 0.1 | 1.5 ± 0.0 | 1.4 ± 0.00 |
| OSc | -0.1 ± 0.2 | -0.1 ± 0.1 | 0.1 ± 0.2 |
| | | | |
| Isoprene (ppb)[a, b] | 1.6 ± 1.9 | 1.5 ± 1.5 | 1.8 ± 1.5 |
| Monoterpenes (ppb)[a, b] | 0.3 ± 0.2 | 0.2 ± 0.1 | 0.2 ± 0.2 |
| Benzene (ppb)[a] | 0.04 ± 0.02 | 0.03 ± 0.01 | 0.05 ± 0.02 |
| NO (ppt)[c] | 24.1 ± 36.4 | 21.5 ± 16.2 | 29.7 ± 29.5 |
| NO$_2$ (ppt)[c] | 593.0 ± 445.8 | 369.4 ± 257.7 | 872.0 ± 478.6 |
| NO$_y$ (ppt)[c] | 934.5 ± 490.0 | 610.5 ± 271.8 | 1283.9 ± 412.6 |
| O$_3$ (ppb)[c] | 24.1 ± 11.9 | 20.1 ± 8.3 | 30.5 ± 13.2 |
| CO (ppt)[c] | 120.6 ± 21.5 | 109.6 ± 12.6 | 134.7 ± 21.7 |
| SO$_2$ (ppb)[b, c] | 31.3 ± 174.2 | 45.0 ± 212.0 | 23.1 ± 64.2 |
| | | | |
| Temperature (°C)[c] | 20.6 ± 4.6 | 20.3 ± 4.5 | 22.4 ± 4.5 |
| Relative Humidity (%)[c] | 73.8 ± 17.3 | 68.6 ± 17.0 | 73.5 ± 17.6 |

[a] VOC measurements using University of Minnesota's PTR-QiToF from the 34-m inlet on the PROPHET tower.
[b] Table entries where the differences between the northerly and southerly backward-trajectories mean values are not
statistically significant (p < 0.01) using a two-sample t-test.
[c] Trace gas and meteorological parameters measured from onboard the MAQL.

## 3.3 Organic aerosol source apportionment

A three-factor solution is obtained for both the above-canopy inlet at 30 meters above the UMBS forest floor (A-MO-
OOA, A-IEPOX-OA, A-91Fac) and the below-canopy inlet at 6 meters within the UMBS canopy (B-MO-OOA, B-IEPOX-
OA, and B-91Fac). The addition of more factors to the PMF solution beyond three factors resulted in less physically
meaningful and interpretable factors. Thus, the three-factor solution is considered the optimal solution for the above- and
below-canopy OA datasets. In this study, seed = 0 was chosen for both PMF solutions as there was minimal variation in
Q/Q$_{exp}$ across the 50 seeds. Solutions at FPEAK = 0 were chosen because solutions at other FPEAK values did not present
improved correlations between factors and reference mass spectra.
Time series of mass concentrations and time series of fractional contributions to total OA are shown in Figure 1. The
O:C ratio of each of the OA factors are as follows: 0.65 (IEPOX-OA), 0.71 (91Fac), and 0.89-0.90 (MO-OOA), all of which
indicate the high degree of oxygenation in each of the factors. For the above-canopy PMF solution, 91Fac makes the largest
contribution to the total OA (43.8%), followed by IEPOX-OA (32.8%) and MO-OOA (23.4%). For the below-canopy PMF





solution, 91Fac also makes the largest contribution to the total OA (42.5%), followed by IEPOX-OA (34.0%) and MO-OOA
(23.5%).  Details on the chemical composition, mass spectral characteristics and diurnal profiles of each factor are discussed
further in the SI.

Hourly averages of the above-canopy OA factors are paired with hourly two-day backward-trajectories for WPSCF
analysis.  Results from WPSCF analysis using above-canopy OA data (Figure 2) indicate that A-MO-OOA predominantly
originates from southerly air masses, as supported by external measurements data such as benzene, OVOCs, carbonyls, and
$SO_4$.  Air masses that originate from the south pass over the large urban centers of Chicago, Milwaukee, and Detroit.  This
further supports the aged, transported nature and anthropogenic influences of the observed A-MO-OOA at this site.  In
contrast, no strong indication of a distinct source region is observed for 91Fac.  This could suggest a more localized source
of 91Fac in relation to the site.  Combining these WPSCF results with correlations of 91Fac with VOC masses corresponding
to monoterpene oxidation suggests that 91Fac is sourced from local biogenic, monoterpene-oxidation related chemistry (Xu
et al., 2018).  Finally, WPSCF results for A-IEPOX-OA indicate that this OA factor coincides with more northerly airflow
with some contributions from northwesterly flow.  These northerly source regions driving A-IEPOX-OA correspond to more
rural, less anthropogenically influenced locations in Canada.  Interestingly, an instance of high WPSCF values for A-
IEPOX-OA and A-MO-OOA can be traced back to areas near the southeastern tip of Missouri and western Tennessee,
implying long-range transport of these OA factors.  Backward-trajectories associated with this  instance of high WPSCF
values for IEPOX-OA pass over areas in the high isoprene-emitting region in the Ozarks of southern Missouri, which is
commonly referred to as the "isoprene volcano" (Carlton and Baker, 2011; Wiedinmyer et al., 2005).  The median mass
concentrations of above-canopy OA factors used as the criterion values for the WPSCF method were A-MO-OOA: 0.31 µg
$m^{-3}$, A-91Fac: 0.68 µg $m^{-3}$, and A-IEPOX-OA: 0.58 µg $m^{-3}$.  Overall, WPSCF analysis indicates that transport from southerly
flow (A-MO-OOA), local sources (A-91Fac), and transport from northerly flow (A-IEPOX-OA) are related to the OA
factors observed at this site.

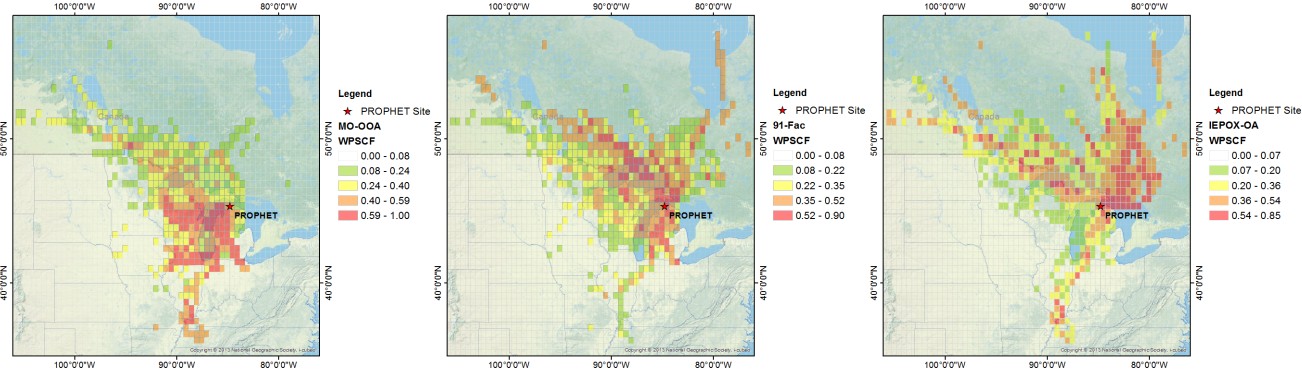

**Figure 2: WPSCF backward-trajectory analysis maps for the following hourly-averaged OA factors from left to right: above-**
**canopy MO-OOA, above-canopy 91Fac and above-canopy IEPOX-OA. The geographical location of the PROPHET site is**
**represented by a red star.  WPSCF analysis is performed using HYSPLIT two-day backward trajectories, a grid cell size of 0.5° by**
**0.5°, and the geographical domain of the extents of the two-day backward trajectories.  Median mass concentrations of above-**
**canopy OA factors (A-MO-OOA: 0.31 µg $m^{-3}$, A-91Fac: 0.68 µg $m^{-3}$, and A-IEPOX-OA: 0.58 µg $m^{-3}$) are used as the criterion**
**WPSCF values.  Color scales correspond to probability of source regions for each respective OA factor, and it should be noted that**
**the range and gradation of the color scale changes between each plot.**
Overall, PMF analysis for both sampling inlets at the site indicates that the OA is a combination of MO-OOA, 91Fac,
and IEPOX-OA.  The dominant OA factor is 91Fac, and during periods of southerly flow MO-OOA contributes relatively
more (time series of the fractional contributions to total OA shown from both inlets is shown in Figure 1 and is also shown
separately for each inlet in Figure S13 and Figure S22 in the SI).  On average, source apportionment results indicate that the



OA at both inlets generally had similar average fractional OA contributions and degrees of oxidation. Diurnal profiles for
above- and below-canopy OA are shown in Figure 3 and indicate that diurnal profiles and variations are similar between the
two inlets for each OA factor. It is worth noting the diurnal pattern of the IEPOX-OA factor, likely indicating a relatively
small influence of local isoprene emissions.

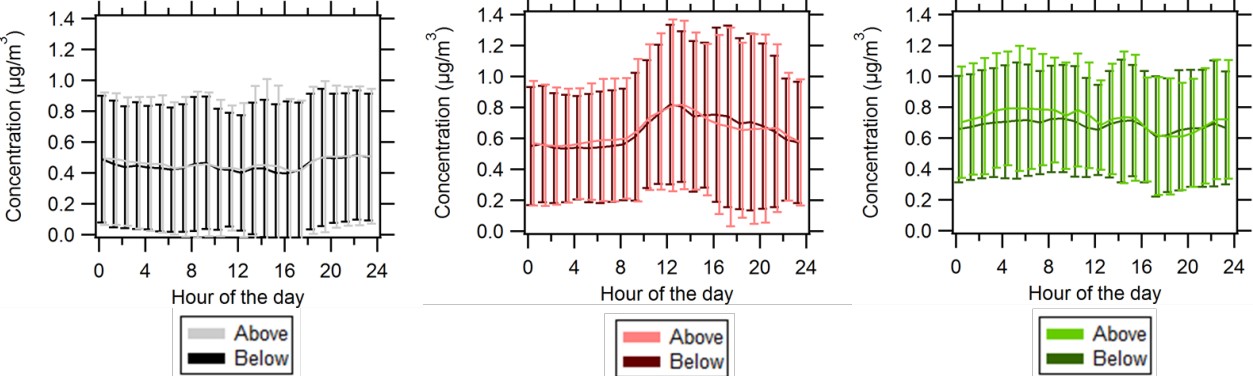

**Figure 3: Diurnal profiles of (left) MO-OOA, (middle) IEPOX-OA, and (right) 91Fac where solid lines represent average values**
**and whiskers represent one standard deviation from the mean. Darker colors represent below-canopy OA (B-MO-OOA, B-**
**IEPOX-OA, B-91Fac) while lighter colors represent above-canopy OA (A-MO-OOA, A-IEPOX-OA, A-91Fac). Data for below-**
**canopy OA factors have been offset by 15 minutes simply to aid plot interpretation.**

**3.4 Vertical characterization of above- and below-canopy NR-PM$_1$ and OA factors**
**3.4.1 Similarity between above- and below-canopy environments**
Mean values of NR-PM$_1$, NR-PM$_1$ species, OA elemental ratios, meteorological parameters, VOCs, and trace gases from
the above- and below-canopy inlets are summarized in Table 2. Each of the mean values for the parameters listed in Table 2
are within one standard deviation of each other for the below- and above-canopy sampling heights. However, results from
Wilcoxon rank-sum tests indicate that the medians for a majority of the above- and below-canopy parameters are
significantly different (p = 0.05). To provide a comparison of the two inlets on the same time scale, scatter plots of 30-
minute averaged values are shown in Figure 4 for OA factors, total OA, SO$_4$, and O:C. For reference, a 1:1 line is shown on
each scatter plot. Values falling within one standard deviation are similar between above and below canopy, while values
deviating farther from the 1:1 line indicate values in which the above- or below-canopy environment had differing
concentrations. Figure 4 illustrates that mass concentrations of OA factors, total OA, and SO$_4$ were similar over a majority
of the campaign and suggests that the above- and below-canopy environments were generally coupled from a PM
perspective. It also appears that total OA, MO-OOA, IEPOX-OA, and 91Fac show increases above canopy relative to below
canopy at the highest end of the range of the measurements.
Results from a recent study from the PROPHET-AMOS 2016 campaign by Millet et al. (2018) indicate that flux of
isoprene and monoterpenes at the canopy-atmosphere boundary represents over half of the net carbon flux, while OVOCs
constitute a majority of the species with detectable VOC fluxes. Overall, the authors report that the observed and modeled
net carbon flux during the campaign was upward (canopy emission) during the campaign in 2016. Additionally, in-canopy
gradients of directly emitted BVOCs, such as isoprene and monoterpenes, indicate patterns that are consistent with their
respective temperature, light, and physical emission dependencies (higher concentrations in-canopy for isoprene and higher
in the lower canopy for monoterpenes). On the other hand, in-canopy gradients of secondary products from BVOC
oxidation, such as acetic acid and glycoaldehyde, indicate patterns consistent with net nighttime uptake and a weak peak






**Table 2: Campaign-averaged values (+/- one standard deviation) of mass concentrations of NR-PM$_1$, VOC and trace gas mixing ratios, and meteorological parameters above and below the canopy.**

| Parameter [a, b] | Above | Below |
|---|---|---|
| NR-PM$_1$ ($\mu$g m$^{-3}$) | 2.4 ± 1.6 | 2.3 ± 1.5 |
| OA ($\mu$g m$^{-3}$) | 1.9 ± 1.0 | 1.8 ± 1.0 |
| SO$_4$ ($\mu$g m$^{-3}$) | 0.3 ± 0.5 | 0.3 ± 0.4 |
| NH$_4$ ($\mu$g m$^{-3}$) | 0.1 ± 0.2 | 0.1 ± 0.2 |
| NO$_3$ ($\mu$g m$^{-3}$) | 0.04 ± 0.04 | 0.04 ± 0.03 |
| Chl ($\mu$g m$^{-3}$) | 0.01 ± 0.00 | 0.01 ± 0.00 |
| O:C | 0.7 ± 0.1 | 0.7 ± 0.1 |
| H:C [c] | 1.5 ± 0.1 | 1.5 ± 0.1 |
| OSc | 0.0 ± 0.2 | -0.1 ± 0.2 |
|  |  |  |
| Isoprene (ppb) | 1.6 ± 1.9 | 1.8 ± 2.2 |
| Monoterpenes (ppb) | 0.3 ± 0.2 | 0.3 ± 0.3 |
| Benzene (ppb) | 0.04 ± 0.02 | 0.04 ± 0.02 |
| NO (ppt) | 36.0 ± 56.9 | 24.1 ± 36.4 |
| NO$_2$ (ppt) | 564.8 ± 393.2 | 593.0 ± 448.8 |
| O$_3$ (ppb) | 32.2 ± 12.2 | 24.1 ± 11.9 |
|  |  |  |
| Temperature (°C) [c] | 20.7 ± 4.1 | 20.6 ± 4.6 |
| Relative Humidity (%) | 71.5 ± 17.3 | 73.8 ± 17.3 |

[a] Summary statistics for NR-PM$_1$ measurements were calculated using 5-minute averaged data, VOC measurements using 1-minute averaged data (University of Minnesota's PTR-QiToF 5-m inlet and 34-m inlet), NO$_x$ above-canopy measurements using 5-minute averaged data (University of Toronto), NO$_x$ and O$_3$ below-canopy measurements using 5-minute averaged data (MAQL), and O$_3$ 1-minute averaged data (CU-Boulder's 27-m inlet on the AmeriFlux Tower).

[b] Percent of data below detection limit for NR-PM$_1$: NH$_4$ (4% / 4%) and Chl (59% / 63%), VOCs: isoprene (30% / 30%), monoterpenes (56% / 36%), benzene (100% / 100%), and NO (37% / 22%), where percentages are shown for above- and below-canopy data, respectively. Unless otherwise stated, the percentage below detection limit for all other parameters is 0%.

[c] Table entries where the null hypothesis ("equal medians") between the above- and below-canopy data cannot be rejected and was not statistically significant ($p > 0.05$) using a two-sided, non-parametric Wilcoxon rank-sum test. All other table entries indicate that the null hypothesis ("equal medians") can be rejected at the statistical significance level ($p < 0.05$).

concentration near the mid-canopy. Correlation analysis of these secondary oxidation products and HCOOH indicates that HCOOH likely originates from a secondary source in this environment (Alwe et al., 2019). In the present study, the overall homogeneity in OA factors implies that, despite vertical gradients in trace gases and BVOCs from primary emission and secondary production, turbulent mixing of aerosols between the forest canopy and the atmosphere is efficient.

Furthermore, this similarity between the above- and below-canopy environments suggests that the chemical timescales of SOA formation processes likely are long relative to residence times due to turbulent mixing (Foken et al., 2012), assuming relatively constant background levels. Ultimately, the observed similarity agrees with previous modeling work that predicts similar SOA mass loadings at these two heights (Ashworth et al., 2015; Schulze et al., 2017). The results shown in the present work also are in agreement with measurements at the site in 2009, showing that aerosol gradients on the PROPHET tower "existed at times between the above-canopy (31.4m) and understory environments (5 m)," but that the understory conditions were generally similar to that of the above-canopy conditions (VanReken et al., 2015). The vertical similarity in





**Figure 4: Scatter plots of above- (A) and below- (B) canopy hourly-averaged values for: (top from left to right) SO$_4$ and total OA, (middle from left to right) MO-OOA and 91Fac, and (bottom) IEPOX-OA and campaign O:C. The three-way valve switched between the two inlets at 10-minute intervals, so 30-minute averaged mass concentrations allows for a comparison of both inlets on the same time basis. Above-canopy values are plotted on the y-axis of each plot and below-canopy values are plotted on the x-axis of each plot. Averages are shown with squares; whiskers represent one standard deviation from the mean. For reference, a 1:1 line is shown with each plot. Outliers on the O:C plot corresponding to periods of precipitation were removed.**


NR-PM$_1$ is also in agreement with findings in the Amazon forest, where a balance between upward and downward fine
particle fluxes was found (Rizzo et al., 2010) and in a southeast Asian rainforest, where PM$_1$ did not show significant
variations with height during the daytime  (Whitehead et al., 2010).
**3.4.2 Episodes of vertical differences in NR-PM$_1$**
Episodes of vertical differences in NR-PM$_1$ between the two inlets were observed, and four such episodes are described
here: Episode #1 (Period: 7/3/2016 19:30 to 7/5/2016 15:00 EDT), Episode #2 (July 11 15:00 to July 12 23:00 EDT),
Episode #3 (7/16/2016 21:30 to 7/19/2016 08:30 EDT), and Episode #4 (7/26/2016 08:30 to 7/31/2016 13:30 EDT).
Episodes were defined as sustained periods in which a vertical difference in OA or SO$_4$ was greater than or equal to
concentrations representing $\geq$ 25% of the campaign averaged OA or SO$_4$.  Episodes #1, #3, and #4 indicate increased above-
canopy OA concentrations, while Episode #2 indicates increased below-canopy SO$_4$ concentrations.  Episodes with higher
above-canopy NR-PM$_1$ ranged up to ~1.0 µg m$^{-3}$ higher in OA and ~0.5 µg m$^{-3}$ higher in SO$_4$ relative to equivalent
above/below mass ($\Delta = 0$).  Figure 5 shows time series of vertical differences in OA factors and SO$_4$ and estimates of friction
velocity (u*) at five different heights (not equal to those for the VOC measurements) on the PROPHET tower over the
campaign.  Calculated from three-dimensional wind velocity data, u* is a function of the shear stress at the surface and is
used in this study as a metric of in-canopy mixing (Equation 5).
The "vertical difference," symbolized as $\Delta$ in Figure 5, is defined as the difference between above- and below-canopy
values, where the positive convention indicates larger concentrations above the canopy.  Episodes of total OA factor vertical
difference ($\sum\Delta$OA Factors) are shown in Figure 6 with corresponding observations of u* and vertical differences in SO$_4$ and
O$_3$.  Figure 6 shows that vertical differences in OA correlate with vertical differences in SO$_4$ for Episodes #1 and #4, while
$\sum\Delta$OA Factors during Episode #3 coincides with higher above-canopy O$_3$ concentrations.  Correlations for Episode #2 are
weaker.  These episodes have different OA factor composition: MO-OOA contributing a larger percentage during Episode
#1 and #2, 91Fac contributing roughly half in Episode #3, and IEPOX-OA contributing more than half of the total OA
vertical difference in Episode #4.
**3.5 The role of mixing and particulate SO$_4$ in vertical differences in OA**
The diurnal profile of vertical difference episodes for total OA, SO$_4$, and OA factors in 2016 indicate there is no clear
temporal pattern observed for any of these species (Figure S26 in the SI).  Maximum vertical differences for SO$_4$, total OA,
and MO-OOA are observed around 3PM.  For total OA, a gradual increase occurs at 9AM before reaching a maximum
vertical difference at 3PM, suggesting that the episodes of higher above-canopy NR-PM$_1$ begin shortly after the most
common observed hours of canopy uncoupling from the site in 2009.  The initiation of these events may also be attributed to
venting of the nocturnal boundary layer as it breaks up in the morning hours, as observed for events of upward particle
number fluxes by Whitehead et al. (2010).
To assess the agreement between the occurrence of episodes and micrometeorological measurements of in-canopy
mixing, episode-specific data for Episodes #2-4 are shown in Figure 6.  During Episode #1, u* data are available only at
36m, so no friction velocity data are shown, preventing a full analysis similar to those performed for the other three episodes.
However, based on similarity in sulfate enhancements, it can be assumed that Episode #1 is somewhat similar to Episode #4.
During Episode #2, based on the backward trajectory cluster analysis shown in Table 1 and the WPSCF results, regional
transport is the likely source of the PM, including the enhancements below canopy.  Prior to this episode, SO$_4$ and MO-OOA
are uniform from below-to-above canopy. The below-canopy enhancement then reaches up to 0.3 µg m$^{-3}$ of SO$_4$.
Downmixing of clean air from aloft could lead to lower concentrations above canopy, meaning that the observed difference
would be caused by decreasing above-canopy concentrations, not increasing below-canopy concentrations.  However,
below-canopy concentrations are observed to increase while above-canopy concentrations decrease.  Without a local, below-



canopy source, this indicates that pollutants from above canopy are mixed into the below-canopy region at a rate faster than they are lost to deposition. Regional pollutants, advected to the site above the boundary layer, can be mixed down to the surface with daytime boundary layer growth, as has been found in previous aircraft campaigns (Berkowitz et al., 1998; Thornberry et al., 2001). Downward transport of air masses from the surrounding region has also been hypothesized to contribute to higher ratios of organic nitrogen to organic carbon ratios in water-soluble aerosols within a forest canopy relative to its forest floor (Miyazaki et al., 2014).

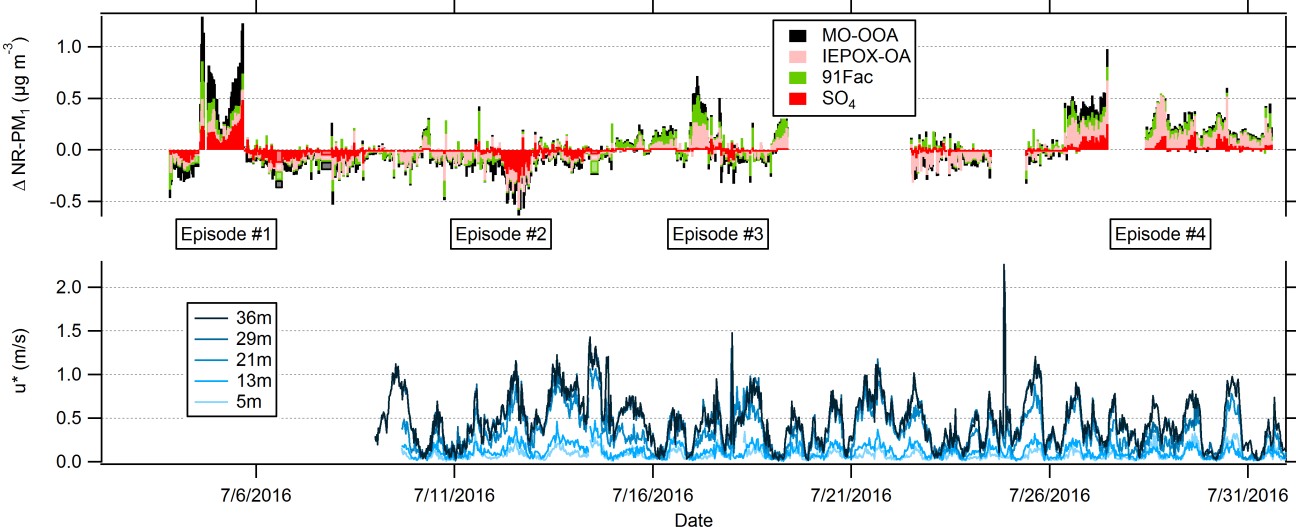

**Figure 5: Time series of (top) observed NR-PM₁ vertical differences between above- and below-canopy inlets and (bottom) friction velocity (u\*) in m/s at five different heights on the PROPHET tower. Vertical differences are defined as Δ Species, where Δ refers to the Above-Canopy minus the Below-Canopy mass concentration. Note that the elevations for u\* are slightly different than those for trace gas sampling.**

During Episodes #3 and #4, periods of above-canopy enhancement in OA factors coincide with relatively lower u\* (u\* < 0.2 m/s during Episode #3 and u\* < 0.6 m/s during Episode #4), with lower friction velocities and greater above-canopy enhancement in Episode 3. In the case of Episode #3, lower in-canopy mixing, mostly occurring during nighttime, agrees well with periods of higher above-canopy OA factor and $O_3$ concentrations. In this case, both IEPOX-OA and 91Fac contribute to the OA enhancement, likely due to strong photochemistry both locally and during transport (as indicated by increased $O_3$). Episode #4 is the longest duration episode; however, the agreement between low-mixing and above-canopy PM enhancements is more variable during this case. Similar to Episode #3, the lowest values of friction velocity occur during the nighttime or early morning. Despite the temporal misalignment of ΔOA and low u\*, it should be noted that friction velocities lower than the campaign average (~0.4 m/s) are observed during the latter periods of Episode #4 (7/28-7/30), which is consistent with the low mixing hypothesis presented for Episode #3. Therefore, it appears that stagnant conditions above the canopy created an environment where canopy exchange became limited and air masses did not fully penetrate into the canopy; aerosol deposition below canopy potentially enhanced the positive delta values. This scenario could also promote in-canopy OA accumulation, but this does not appear to have occurred, implying that other factors contributed to these vertical differences. Instead, it appears that increased sulfate loading was associated with this transport and that this increased sulfate above the canopy led to enhancements above canopy, particularly of IEPOX-OA relative to 91Fac, as shown in Figure 6. This also could be related to associated changes in aerosol liquid water.

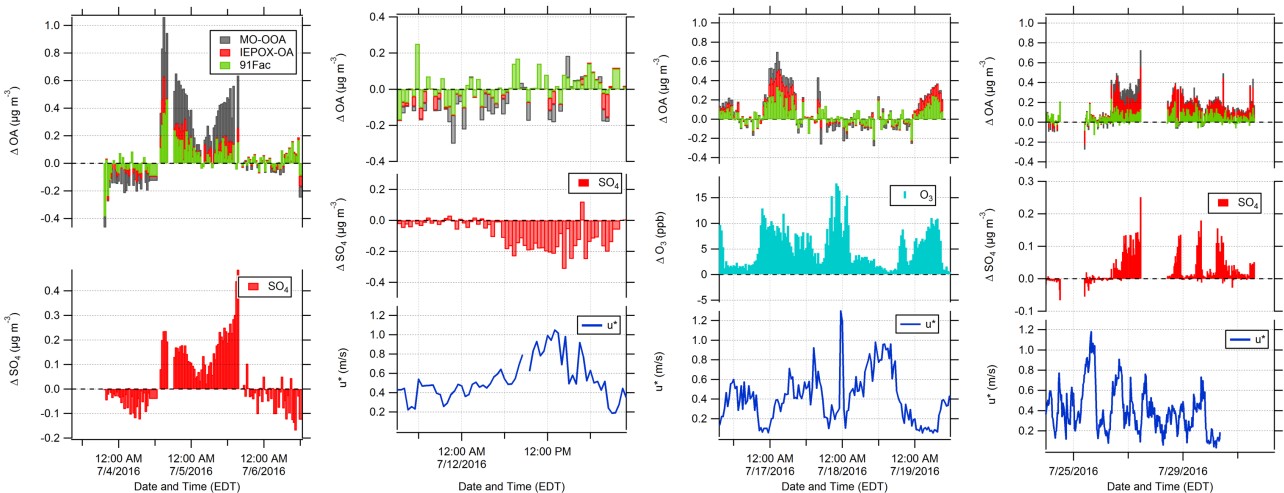

**Figure 6: Time series of observed NR-PM₁ vertical difference episodes: (from left to right) Episode #1, Episode #2, Episode #3, and Episode #4.  Data shown for Episode #1, Episode #2, and Episode #4 include vertical differences in 30-minute averaged above- and below-canopy SO₄, while data shown for Episode #3 show vertical differences in 30-minute averaged above- and below-canopy O₃.  The O₃ data were measured from the AmeriFlux Tower at 6 meters and 27 meters and were provided courtesy of CU-Boulder.  Friction velocity measurements at 29m on the PROPHET tower are also provided for Episodes #2-4.**

The probability distributions of $\Delta SO_4$ and $\Delta O_3$ between episodes and the average for the entire PROPHET campaign are compared in Figure S27 of the SI.  The cumulative distribution function (CDF) for $\Delta SO_4$ indicates that Episodes #1 and #4 had higher probabilities of positive $\Delta SO_4$ (denoting higher above-canopy concentrations), while Episode #2 had higher probabilities of negative $\Delta SO_4$ (denoting higher below-canopy concentrations).  The episode-specific CDFs for #1 and #2 are also different from the total campaign CDF, implying that the vertical differences observed were distinct events.  The average $O_3$ difference between the 6- and 27-m inlets on the AmeriFlux tower throughout the campaign indicated that $O_3$ was on average 5 ppb greater at the 27-m inlet, with the difference reaching as high as 34 ppb on July 6 and July 20.  The probability distribution of vertical $O_3$ differences observed during Episode #3 is markedly different than those observed during Episode #1, Episode #4, and the remainder of the campaign.  This indicates that $O_3$ had a larger probability of being between 0-10 ppb higher above the canopy during this episode compared to the rest of the campaign.

Based on Episode #1 and Episode #4, a potential driving factor of vertical differences for OA is a vertical difference in particulate $SO_4$.  Figure 7 indicates the vertical differences in OA and $SO_4$ relative to their respective above-canopy concentrations.  This metric is defined as "% PM Difference," where PM is the given PM constituent and is equal to the $\Delta PM$ constituent divided by the above-canopy PM concentration.  Data that fall in the upper right and bottom left quadrants correspond to higher above- and higher below-canopy concentrations, respectively.  The vertical difference above or below the canopy can be up to approximately 35% of the total available $SO_4$ or OA at the site.  Figure 7 indicates that there is a strong linear relationship between the %$SO_4$ Difference and %OA Difference during the campaign, even outside of the Episodes defined here.

The concurrent features of vertical differences in OA factors and $SO_4$ observed in Episodes #1, #2, and #4 and OA factors and $O_3$ in Episode #3 suggest that long-range regional transport (enhanced MO-OOA, $SO_4$, and/or $O_3$) is a key first step in causing these differences.  Local through-canopy mixing then determines whether the enhancement is above (Episodes #1, #3, and #4) or below (Episode #2) the canopy.  The transported material can then impact local chemistry.  Increased $SO_4$ likely increases aerosol hygroscopicity resulting in increased ALW, which could result in increased partitioning of semi-volatile organics to the condensed phase (i.e., IEPOX-OA) (Budisulistiorini et al., 2017; El-Sayed et al.,

2018; Marais et al., 2016).  Pre-existing OA (i.e., MO-OOA) transported with SO₄ aerosol could also play a role in the
partitioning of OA above the canopy during these episodes.  Vertical gradients in O₃ coinciding with Episode #3 (higher O₃
above) could initiate relatively local formation of extremely low volatility compounds, which are first-generation oxidation
products of α-pinene and O₃ (Jokinen et al., 2015).  Based on the WPSCF and backward trajectory clustering results, cooler,
northerly air masses associated with IEPOX-OA also could promote partitioning of semi-volatile organics to the particle
phase.  It is important to note that  it is difficult to conclude the exact chemical mechanisms that are influencing these events
due to their low mass loadings and episodic nature and the consistently higher above-canopy O₃ concentrations during the
campaign.



**Figure 7: Scatter plot of % difference in SO₄ (y-axis) and OA (x-axis) for the entire campaign.  The % difference is**
**calculated as Δ Species / Above-Canopy Species concentration, and is representative of the normalized Δ Species**
**concentration.**
Understanding these events is important because uncoupled forest-canopy conditions have been observed in a number of
locations (Foken et al., 2012; Whitehead et al., 2010) and could indicate differences between above-canopy and surface level
PM.  One-dimensional modeling has revealed that the timescales of turbulent transport inside a forest canopy can be much
shorter (minutes) than the timescale of aerosol dynamics and deposition (hours) (Rannik et al., 2016), but this evaluation
suggests that advective episodes can cause vertical gradients that are not locally driven.  Above-canopy OA and particle
fluxes have been observed in other studies (Farmer et al., 2013; Pryor et al., 2007), suggesting the need for careful evaluation
of whether differences are driven by local chemistry, long-range transport, or a combination thereof.

**3.6 Particle dry deposition model**

To investigate broadly the effects of canopy mixing on particle deposition in a forest canopy, a particle dry deposition
model is used.  Note that we are not applying this through the canopy but to illustrate the relationship between deposition
and u*.  The resistance model for particle dry deposition assumes that the deposition process is controlled by three
resistances in series: aerodynamic, quasi-laminar, and canopy resistance (Seinfeld and Pandis, 2006).  Details on the particle
dry deposition model are included in the SI.  Conditions and parameters representative of the land use category, season, and
forest canopy present at the PROPHET site are used as model inputs.





Figure 8 displays model results for deposition velocity as a function of particle diameter under stable atmospheric
conditions. For illustration, test cases are shown for u* values for low in-canopy mixing (u* = 0.1 m/s) and high in-canopy
mixing (u* = 0.8 m/s). Values of u* were chosen based on the range of u* values observed during the campaign at 29m on
the PROPHET tower. For the submicron particle diameter range (0.1 to 1 micron), a characteristic minimum in deposition
velocity is observed for both test cases at 1 micron diameter. Deposition modeling indicates that higher deposition velocities
are achieved in the high in-canopy mixing test case, implying that there is less canopy transport resistance to the surface and
more deposition in the canopy. This corresponds to Episode #2.

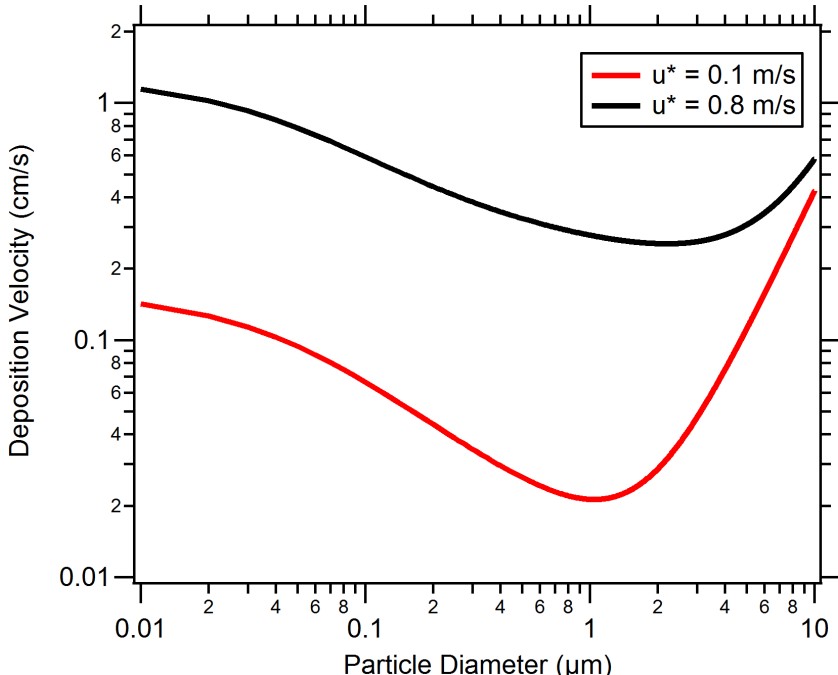

**Figure 8: Particle dry deposition velocity resistance model results plotted by particle diameter and friction velocity.**
**Two test cases of low and high friction velocity, representative of data collected at the PROPHET site, are plotted in**
**red and black solid lines, respectively. Land-use conditions and seasonal parameters similar to the PROPHET**
**campaign were used as inputs to the particle dry deposition model. Additional model parameters are provided in the**
**SI.**
Deposition modeling also indicates relatively lower deposition velocities for the low mixing condition test case, which
suggests that particles aloft are transported less efficiently into the forest (similar to the conditions present for Episodes #3
and #4). A comparison of the deposition resistances (shown in Figure 9) indicates that aerodynamic and quasi-laminar
resistances in the low-mixing test case are an order of magnitude greater than the high-mixing case. The settling removal
resistance is the same between both test cases because it is merely a function of diameter. A greater aerodynamic resistance
limits turbulent transport from the above-canopy layer to the surface layer and greater quasi-laminar layer resistance limits
transport to just above the surface by lowering particle impaction. In total, this result emphasizes the importance of in-
canopy transport and mixing in governing particle concentrations in forests.



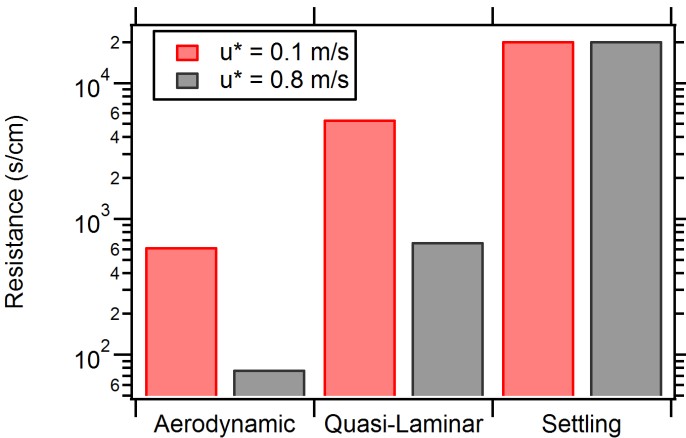

**Figure 8: Deposition and particle settling (accounts for the effect of sedimentation) resistances comparison for low**
**and high friction velocities. Resistance comparison assumes a 1 micron particle diameter.**
**4 Conclusions**

710   In this study, source apportionment of OA using separate PMF analyses on inlets situated above and below a forest
canopy resulted in three OA factors at a site in northern Michigan. Similarity in OA composition, concentration, and diurnal
profiles was observed between the two inlets, suggesting that turbulent transport efficiently mixes OA across the canopy.
However, OA factor vertical differences between the two inlets were observed during four separate episodes. During these
episodes, vertical differences were both positive (greater concentrations above the canopy) and negative (greater
concentrations below canopy). NR-PM$_1$ composition amongst these episodes was unique (Episode #1: MO-OOA dominant,
Episode #2: SO$_4$ dominant, Episode #3: 91Fac dominant, Episode #4: IEPOX-OA dominant). Furthermore, the vertical
difference concentrations can be 35% of the total available SO$_4$ or OA at the site. It has also been shown that relative
vertical differences in SO$_4$ and OA are linearly associated, suggesting the role of regional SO$_4$ in vertical profiles of NR-PM$_1$
in forested environments.
720   Using micrometeorological measurements, it has been hypothesized that periods of low in-canopy mixing
conditions can be associated with the timing of vertical difference episodes where above-canopy concentrations are larger
than those below. The opposite scenario is hypothesized to be associated with transport of air from aloft or with
accumulation of material below canopy more rapidly than it is deposited. Either way, these results suggest that canopy
mixing impacts particle levels. However, under low-mixing scenarios, it appears that enhancements as defined here depend
on both these conditions and the presence of transported O$_3$ or SO$_4$ to enhance IEPOX-OA and 91Fac levels.
726   To the knowledge of the author, this is one of only a few studies that have assessed vertical profiles in OA factors
above and below a forest canopy, and the first study to observe episodes of vertical differences in OA factors above and
below a forest canopy. This work adds to the existing literature on aerosol chemistry in a forest canopy environment
presented by Rizzo et al. (2010) and Whitehead et al. (2010) and to literature on vertical profiles of OA in an urban
environment by Özturk et al. (2013).
731   To investigate the effects of forest canopies on SOA formation, small-scale models, such as those described in
Schulze et al. (2017) and Ashworth et al. (2015), have been developed. The OA data from this work can be used to validate
such models and are particularly relevant to these modeling efforts, as the models were developed using campaign data from
the PROPHET site in 2009. Vertical transport and horizontal advection are both sources of uncertainty in current forest
canopy-atmosphere exchange models, which are designed to focus on local processes. Ultimately, the vertical similarity in



NR-PM$_1$ OA composition observed in this study implies that it may be valid to assume that below-canopy OA composition
is generally representative of the OA composition in the atmospheric layer directly above the canopy and vice versa.
However, this work highlights that advection of regional pollution into forested regions can lead to in-canopy gradients that
are not present under purely local conditions.

**Data Availability**

Data are available through contacting the corresponding author.

**Author Contributions**

A.A.T.B. prepared the manuscript with input from all authors. A.A.T.B., H.W.W. and R.J.G. conceived of the study and
operated and analyzed the data from the HR-ToF-AMS. S.K. and A.L.S. investigated friction velocity and in-canopy
mixing. H.D.A. and D.B.M. collected and analyzed VOC data. J.H.F., M.H.E. and S.A. collected and analyzed trace gas
data and operated the MAQL.

**Competing Interests**

The authors declare they have no conflict of interest.

**Acknowledgements**

The assistance of all staff and collaborators at UMBS is gratefully acknowledged. We would like to thank Wei Wang at
University of Colorado Boulder for assistance in collecting O$_3$ data at the AmeriFlux tower and the Murphy group at the
University of Toronto for provision of nitrogen oxide data. This work was funded by the National Science Foundation
(NSF) under grant AGS-1552086. PTR-QiToF measurements during PROPHET-AMOS were also supported by NSF (grants
AGS-1428257+AGS-1148951). Trace gas and meteorological measurements on board the MAQL were also supported by
NSF (grant AGS-1552077).

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

Pittsburgh: insights into sources and processes of organic aerosols. Atmospheric Chemistry and Physics 5, 3289–
3311.