# Peer review of "Transport-driven aerosol differences above and below the canopy of a mixed deciduous forest"

_Atmospheric Chemistry and Physics, 2021_

## Referee Comment (RC1)

The authors present aerosol measurements in a mixed deciduous forest above and below the canopy and more importantly include both horizontal and vertical transport mechanisms. They report similarity in the OA concentrations above and below canopy but also identify 4 different episodes, during which above and below canopy OA concentrations are different. PMF analysis is performed at two different levels, providing further insights into sources of OA. The manuscript is well written, the method and its limitations are thoroughly discussed, and the presented results are original. The manuscript is a very important contribution to the field, there are very limited studies reporting such analysis. Therefore, it should be published in ACP after minor revisions:

1) In the abstract, the author report that the differences in vertical concentrations reached up to 0.8 μg m-3. It would be useful for the reader to have the same information presented in terms of their percentages too.

2) In the introduction, the author discusses coupling conditions *"In this study, the degree of coupling was calculated using the ratio of the kinematic heat flux above the canopy to the kinematic heat flux in the upper canopy. Opposing kinematic heat flux directions (negative ratios) imply that the below- canopy environment is uncoupled from the atmosphere. In 2009, coupling conditions ranged between strong coupling, weak coupling, and uncoupled."* Could you please add kinematic heat flux criteria used to separate between strong, weak and uncoupled conditions?

3) I really appreciate the author adding theoretical details of PMF analysis in section 2.3, however, the manuscript may read better if that was added in the supplementary document, where details/results of each PMF analysis are discussed.

4) In the caption of Figure 1, could you please explain the sampling of the OA time series presented in Figure 1 by adding that OA data are from both heights (6m and 30m), sampling switched measurement every 10 minutes between the two heights

5) In section 3.4.2, in line 561, the symbol,"Δ", is used, without being defined, however, the author then explains the meaning of the symbol in the next paragraph on line 565. Could you please put those in the right order?

6) Line 567, the author says, "episodes of total OA factor vertical difference (∑Δ OA) are shown in Figure 6". Could you please clarify the total of what? Is that total vertical difference for every 30 minutes?

7) Typo in the supplement: for the caption of Figure S26: instead of "dial" it should be "Diurnal"

8) In section 3.5, line 604, the author states, *"During Episodes #3 and #4, periods of above-canopy enhancement in OA factors coincide with relatively lower u* (u* <0.2 m/s during Episode #3 and u* < 0.6 m/s during Episode #4)"* Could you please clarify what u* values are referred here? Are those based on averaged values during a specific episode? Also, *u*$ from which height?

9) A technical point, In Figure 5, it is not clear exactly which period of the time series are part of the individual episodes. It will be easier if you could you add vertical lines/ shaded area in those figures for those periods.

---

## Author Response (AR1)

**Review 1**

1) In the abstract, the author report that the differences in vertical concentrations reached up to 0.8  $\mu$ g m-3. It would be useful for the reader to have the same information presented in terms of their percentages too.

This information is included in the updated abstract (compared to the campaign average OA concentration of 1.9  $\mu$ g m-3).

2) In the introduction, the author discusses coupling conditions "In this study, the degree of coupling was calculated using the ratio of the kinematic heat flux above the canopy to the kinematic heat flux in the upper canopy. Opposing kinematic heat flux directions (negative ratios) imply that the below- canopy environment is uncoupled from the atmosphere. In 2009, coupling conditions ranged between strong coupling, weak coupling, and uncoupled." Could you please add kinematic heat flux criteria used to separate between strong, weak and uncoupled conditions?

It is important to note that the present study did not quantify these ratios or determine coupling conditions as was done in Steiner et al., 2011. However, the text has been updated to reflect values used by Steiner et al., 2011.

"In 2009, coupling conditions ranged between strong coupling (greater than zero but less than the threshold value defined by the slope of a regression between the two heat fluxes), weak coupling (greater than the defined threshold value), and uncoupled (negative)."

3) I really appreciate the author adding theoretical details of PMF analysis in section 2.3, however, the manuscript may read better if that was added in the supplementary document, where details/results of each PMF analysis are discussed.

This has been done in the updated manuscript (changes are not included here, as the changes are essentially a cut of ~6 paragraphs from the manuscript that are now pasted into the supplemental information).

4) In the caption of Figure 1, could you please explain the sampling of the OA time series presented in Figure 1 by adding that OA data are from both heights (6m and 30m), sampling switched measurement every 10 minutes between the two heights.

The following has been added to the end of the caption of Figure 1: "The OA time series include data from both heights (6 m and 30 m), with the sampling switched at 10-minute intervals."

5) In section 3.4.2, in line 561, the symbol, " $\Delta$ ", is used, without being defined, however, the author then explains the meaning of the symbol in the next paragraph on line 565. Could you please put those in the right order?

The following text has been moved upward by several lines to clarify: "The "vertical difference," symbolized as  $\Delta$  in Figure 5, is defined as the difference between aboveand below-canopy values, where the positive convention indicates larger concentrations above the canopy".

6) Line 567, the author says, "episodes of total OA factor vertical difference ( $\Sigma\Delta$  OA) are shown in Figure 6". Could you please clarify the total of what? Is that total vertical difference for every 30 minutes?

The 'total' refers to the sum of the delta values for each of the three factors. The text has been updated to the following: "The sum of the delta values for each OA factor ( $\sum \Delta OA$  Factors) for the episodes are shown in Figure 6..."

7) Typo in the supplement: for the caption of Figure S26: instead of "dial" it should be "Diurnal"

This has been corrected.

8) In section 3.5, line 604, the author states, "During Episodes #3 and #4, periods of above-canopy enhancement in OA factors coincide with relatively lower u\* (u\* <0.2 m/s during Episode #3 and u\* < 0.6 m/s during Episode #4)" Could you please clarify what u\* values are referred here? Are those based on averaged values during a specific episode? Also, u\* from which height?

Here, we have added a statement that indicates the height and temporal duration used for this assessment (average of all heights and over the duration of the episode).

9) A technical point, In Figure 5, it is not clear exactly which period of the time series are part of the individual episodes. It will be easier if you could you add vertical lines/ shaded area in those figures for those periods.

Figure 5 has been updated with vertical lines/shading to indicate these time periods.

**Review 2**

1) lines 123-128: I think the authors could sharpen the stated objectives of this paper. The paper presents a few cases that have some scientific, and performed analysis has probably a scientific value by itself. Just providing potentially useful data for future modeling activities is not convincing for a research paper.

Here, the following text has been added. "with the aim of evaluating quantitatively potential chemical and physical phenomena leading to observed differences between particulate matter above and below the forest canopy."

2) The different OA factors, first discussed in section 3.3, should be spelled out in the main text.

MO-OOA and IEPOX-OA are defined in section 2.3 (and have not been moved to the supplemental information in response to a comment from Reviewer 1). We have added a statement in section 3.3 to define 91-Fac as a factor with an enhanced signal at m/z 91 in the mass spectrum.

3) lines 568-572: One can see qualitatively from Figure 6 whether certain compounds appear to follow each other directly or inversely. However, plotting time series does not tell anything about correlations without further statistical analysis.

The text has been updated to include correlation coefficients between various parameters for each episode:

Episode #1  $\sum \Delta OA$  Factors-Sulfate, r = 0.69 Episode #2 Sulfate-u\*, r = -0.44 Episode #3  $\sum \Delta OA$  Factors-Ozone, r = 0.34 Episode #4  $\sum \Delta OA$  Factors-Sulfate, r = 0.59

4) lines 574-575: Diurnal profiles reveal only diurnal patterns and do not tell anything about temporal patterns in some other time scales.

The word temporal has been changed to diurnal in the text at this location.

5) The statement on lines 639-640 is rather strong, especially as many other potential reasons for the apparently strong relation seen in Figure 7 (as discussed separately on lines 647-660).

To decrease the strength of this statement, we have changed "a potential driving force is" to "one of several potential driving forces is".

6) line 699: Figure 9 should be Figure 8.

We believe Figure 9 is correct. Figure 8 shows the deposition velocities, while Figure 9 shows the resistances that contribute to the deposition velocities. No changes have been made in response to this comment.

7) The way the second paragraph in section 4 (lines 720-726) has been written (... has been hypothesized) makes it difficult for a reader to understand whether this paragraph presents general information or results from the current work.

The word hypothesized has been changed to 'shown in this work' in this paragraph.

8) Please check out how to express days of the year and times of the day in the text.

Based on the ACP guidelines, within the text, dates now follow the format day month year and times of the day now follow the format hour : minute LT (where LT refers to local time, which was Eastern Daylight Time).